# Usability of Two New Interactive Game Sensor-Based Hand Training Devices in Parkinson’s Disease

**DOI:** 10.3390/s22166278

**Published:** 2022-08-21

**Authors:** Lea Saric, Samuel E. J. Knobel, Manuela Pastore-Wapp, Tobias Nef, Fred W. Mast, Tim Vanbellingen

**Affiliations:** 1Department of Psychology, University of Bern, 3012 Bern, Switzerland; 2Gerontechnology and Rehabilitation Group, ARTORG Center for Biomedical Engineering Research, University of Bern, 3008 Bern, Switzerland; 3Neurocenter, Luzerner Kantonsspital, 6000 Luzern, Switzerland

**Keywords:** exergaming, sensor, dexterity, Parkinson’s Disease, flow, usability

## Abstract

This pilot cross-sectional study aimed to evaluate the usability of two new interactive game sensor-based hand devices (GripAble and Smart Sensor Egg) in both healthy adults as well as in persons with Parkinson’s Disease (PD). Eight healthy adults and eight persons with PD participated in this study. Besides a standardised usability measure, the state of flow after one training session and the effect of cognitive abilities on flow were evaluated. High system usability scores (SUS) were obtained both in healthy participants (72.5, IQR = 64.375–90, GripAble) as well as persons with PD (77.5, IQR = 70–80.625, GripAble; 77.5, IQR = 75–82.5, Smart Sensor Egg). Similarly, high FSSOT scores were achieved after one training session (42.5, IQR = 39.75–50, GripAble; 50, IQR = 47–50, Smart Sensor Egg; maximum score 55). Across both groups, FSSOT scores correlated significantly with SUS scores (r = 0.52, *p* = 0.039). Finally, MoCA did not correlate significantly with FSSOT scores (r = 0.02, *p* = 0.9). The present study shows high usability for both interactive game sensor-based hand training devices, for persons with PD and healthy participants.

## 1. Introduction

Parkinson’s Disease (PD) is a progressive neurodegenerative condition that leads to impairments in motor control and deficits in nonmotor cognitive tasks [1]. The disease is characterised by its cardinal symptoms bradykinesia, rigidity, tremor, and, in later stages, postural instability [2]. Cognitive impairments, such as deficits in working memory, planning, and attention, are commonly observed as the disease progresses [3,4,5]. Deficits in manual dexterity are often present in early stages [6,7,8]. Patients are given medication to relieve general motor symptoms [9]. However, medication, while improving cardinal symptoms [10], appears to have a negligible effect on impaired dexterity [7] and may even impair cognitive functions [11]. Complementary forms of conventional treatment, such as occupational or physical therapy interventions, have been used successfully to help patients engage in activities of daily living (ADL) [12], also showing improvement in dexterity-related ADL with training [13]. However, the evidence for long-term improvements in dexterity after training, particularly in persons with PD, is still limited [14]. Furthermore, conventional rehabilitation procedures are rather repetitive and monotonous, and patients often lack the necessary motivation [15].

A rehabilitation tool that has gained increasing attention in recent years for its potential to improve exercise adherence is exergames [16,17]. Exergames are a combination of physical exercise and video gaming, which require the player to move the body or body parts in order to maintain online control of the avatar (the player’s virtual shape) displayed on screen. The interactive training modalities are often highly motivating and fun, and users sometimes lose track of time during training, getting into a flow experience. If the latter is the case, this might further improve the outcome of training [18]. Additionally, exergames require cognitive processing [19], and it has been suggested that the precise nature of the interaction between physical and cognitive components is the reason for their beneficial effects [20,21]. The majority of studies using exergames provide promising results as they were able to demonstrate improvements of balance, motor skills, and cognitive functions in persons with PD [22,23,24]. Some studies also evaluated the effects of exergame-based training to improve hand function by means of the Leap Motion Controller, which is an optoelectronic device capable of tracking hand and finger movements [25]. However, this type of device does not allow any tactile pressured feedback.

The aim of this study is to assess the usability of a new interactive game sensor-based hand training for both healthy participants and persons with PD. Unlike previous devices, these new training devices feature pressure sensors instead of only tracking the visual feedback of movements [25,26]. The resulting tactile pressured feedback could lead to a more precise training of hand and finger strength. We hypothesise a high usability, due to the interactive and motivating nature of the new training devices. We further hypothesise that high usability induces more flow experience. Finally, we were interested how mild cognitive impairment may affect flow and usability.

## 2. Materials and Methods

### 2.1. Participants

For this cross-sectional study, both persons with PD as well as healthy participants were recruited. All participants were recruited from the Luzerner Kantonsspital. Persons with PD were included if they met the following criteria: (1) PD diagnosis defined by the UK Parkinson’s Disease Society Brain Bank Criteria [27], (2) Hoehn and Yahr stage between I and III, (3) aged ≥55 and ≤80, and (4) reported subjective dexterous difficulties during clinic visits. Exclusion criteria included (1) severe medical conditions, including psychiatric disease, (2) impaired cognitive functioning (MoCA score < 21), (3) excessive or uncontrollable tremors of the upper extremities, or (4) presence of any neurological disorder other than PD. We tried to include age- and sex-matched healthy participants. Written informed consent was obtained from all participants according to the latest Declaration of Helsinki. Ethical approval was granted by the Ethics Committee of Nordwest- und Zentralschweiz, Switzerland. The trial is registered at the RAPS (Registry of all Projects in Switzerland), Swissethics (BASEC ID 2019-00433). The study conformed to the STROBE guidelines for cross-sectional studies (https://www.equator-network.org/reporting-guidelines/strobe/, accessed on 12 July 2022).

### 2.2. Materials

The System Usability Scale (SUS) [28] was used to evaluate the new game sensor-based training devices. The SUS is a well-validated questionnaire, consisting of a 10-item, 5-point Likert scale, and each item is scored from 0 (*strongly disagree*) to 4 (*strongly agree*). Three usability criteria are taken into account: effectiveness, satisfaction, and efficiency. The total score is obtained by multiplying the mean sum value by 2.5. The SUS score ranges from 0% to 100%, where a higher score indicates better system usability. A score of 70% up to a maximum of 100% represents acceptable-to-excellent usability [29]. An adapted version of the Flow State Scale for Occupational Tasks (FSSOTs) [30] was used to evaluate the flow state, which comprehensively assesses the experienced flow during an activity [31]. The questionnaire includes 11 items, each of which was rated on a 5-point scale 1 (strongly disagree) to 5 (strongly agree).

The Montreal Cognitive Assessment (MoCA) [32] was used to assess cognitive function. The MoCA is divided into short-term memory, visuospatial abilities, multiple aspects of executive function, working memory, language, and orientation to time and place. The maximum score is 30; a score below 26 would indicate mild cognitive impairment in PD.

### 2.3. Procedure

Two interactive hand training devices GripAble [33] and the Smart Sensor Egg were used for the sensor game-based training. The training session took in total around 25 min. All participants (persons with PD and healthy participants) tested the GripAble; five additional persons with PD used the Smart Sensor Egg.

The GripAble (https://gripable.co/, accessed on 12 July 2022) (see Figure 1) is a new wireless device connected (by Bluetooth) with a tablet on which an app including different therapy games is installed. The device can capture fine hand and finger movements. GripAble allows the training of different wrist and hand movements (wrist extension/flexion, radial/ulnar deviation, and pronation/supination) and grip and pinch forces. 

So far, nine games are available, of which four were chosen for the present study. Each of these four games focuses on different hand/finger movements.

*Balloon Buddies* requires a controlled grip (hand/finger strength and endurance) and controlled release. The goal is to control an owl by varying the pressure on the device (smooth transition between grip and release) (see Figure 1). Squeezing GripAble inflates the balloon attached to the owl to make the owl move up on the screen. Releasing GripAble brings the owl down. The owl needs to collect all the stars to gain points. Different levels allow for sufficient time and complexity of the required control. Depending on the level of difficulty, the stars appear on different spots on the screen during gameplay (see Figure 2 top left).

*Windowsill* focuses on controlled pronation and supination and grip release. This activity presents pots in different places on a windowsill. A bag of soil is then moved from left to right (using pronation/supination) until it is placed directly above one of the pots. When stable over the pot, the soil can be released to fill it by gripping. Afterwards, a seed can be placed into the pot. This is followed by a watering can, which needs to be poured until the flower appears. As the levels progress, more and smaller pots appear (see Figure 2 top right).

*Concierge* uses wrist extension/flexion and grip forces to control a hotel elevator (lift in English) to deliver objects and people to the right floor as quickly as possible. Every object and every group of person has a specific colour: For example, hotel guests are green and need delivering to a green door. In this game, the player must hold the GripAble horizontally (see Figure 2 bottom left).

*Pufferfish* focuses on wrist ulnar and radial hand deviation and, in higher levels, on grip and release. The user controls the fish to move up and down the screen by moving GripAble through wrist radial and ulnar deviation. The goal is to collect bubbles. From Level 3 onwards, the complexity increases with more items falling into the water, which need to be avoided or can be blown away by squeezing (see Figure 2 bottom right).

The second device that was used is the *Smart Sensor Egg*; it is a novel device created for hand training (see Figure 3).

Beside accelerometer and gyroscope, the device includes a pressure sensor. Like the GripAble, one can connect Smart Sensor Egg via Bluetooth with a tablet or smartphone. We used the self-developed game Asteroid Shooter game.

This is a shooter game in which the player has control over a spaceship (see Figure 4). The main goal is to shoot all the targets that approach the ship. The orientation of the Smart Sensor Egg is translated into the position of the spaceship on screen. The spaceship can move left and right, depending on the supination or pronation of the wrist. The targets can be shot by moving the spaceship to the correct position (directly under the approaching target) and then by squeezing the Smart Sensor Egg. The game includes ten difficulty levels. The difficulty changes with the numbers of targets, the speed of the targets, and the presence of distractors that are not supposed to be shot.

### 2.4. Statistical Analyses

The normality of data was established using the Shapiro–Wilk test. However, due to the small sample sizes, nonparametric statistics were used for all measures. Equivalency between groups in regard to their demographic and clinical characteristics was evaluated by means of the Mann–Whitney test or the chi-square test. Further between-group analyses concerning the usability and flow scores were performed using the Mann–Whitney test. To check for within-group differences between the ratings of the devices, the paired samples Wilcoxon test was used. Spearman or eta correlations were performed in order to explore relationships between outcome measures as well as demographic characteristics. The level of significance was set at *p ≤* 0.05 (two-tailed). The data were analysed in R (version 4.0.5) [34] as well as in IBM27 (IBM Corp. Released 2020. IBM SPSS Statistics for Windows, Version 27.0. Armonk, NY, USA: IBM Corp).

## 3. Results

A total of eight persons with PD as well as eight age-matched healthy participants were recruited (*Mdn* = 63, *IQR* = 58–68.25). In regard to demographic characteristics, there were no significant differences between the groups except for gender (*χ**^2^* = 4, *p* = 0.046). Detailed clinical and demographic characteristics are presented in Table 1.

The median SUS scores obtained after playing with the GripAble were 72.5 (*IQR* = 64.375–90) for the healthy participants and 77.5 (*IQR* = 70–80.625) for the persons with PD, indicating acceptable usability. There was no significant difference between these scores (Mann–Whitney test, *p* = 0.83). The overall SUS score of the GripAble was 77.5 (*IQR* = 64.375–83.75). Furthermore, the analysis revealed no significant differences regarding the usability ratings of the GripAble (*Mdn* = 77.5; *IQR* = 70–80.625) and the Smart Egg (*Mdn* = 77.5; *IQR* = 75–82.5) in the PD group (Wilcoxon test, *p* = 0.28). 

The FSSOTs revealed high flow scores both in the healthy participants (*Mdn* = 42; *IQR* = 39.25–50.75) as well as in the persons with PD (*Mdn* = 43.5; *IQR =* 40.5–47.75). No significant difference between these scores (Mann–Whitney test, *p* = 0.96) was found. There is a trend for significance for the Smart Sensor Egg (*Mdn* = 50; *IQR* = 47–50) to induce a higher flow than the GripAble (*Mdn* = 43.5; *IQR* = 40.5–47.75) (Wilcoxon test, *p* = 0.058).

We found a significant correlation between SUS and flow scores (*r* = 52, *p* = 0.039, see also Figure 5).

## 4. Discussion

The main aim of this study was to investigate the usability of two new game sensor-based hand training devices, both in healthy participants as well as in persons with PD. Furthermore, we evaluated whether mild cognitive deficits in PD could affect flow and usability. Overall, we found that both healthy participants as well as persons with PD rated both new devices as usable tools. No differences were found between the groups. These findings are in accordance with similar studies, which investigated the usability of exergaming devices for persons with PD [25,35]. The novelty of this usability study is that the two now-new devices have pressure sensors. Connected by Bluetooth with tablets, different games allow now a highly interactive, motivating, and fun training modality, where hand/finger strength (due to the pressure sensor) can be trained. This has not been the case in previous hand exergaming in PD [25,26]. Indeed, previous studies lacked tactile pressured feedback, and they were restricted to tracking visual feedback of the movements [25,26]. It is well known that persons with PD demonstrate a weaker [36] and less precise grip [37], which in turn may lead to disturbed object manipulation [38]. Both GripAble and Smart Sensor Egg provide now a possible new training modality for PD to train and target more precisely hand and finger strength. This study focused on the usability testing of these devices by means of a cross-sectional design, and therefore we cannot make any claims about whether the new hand training devices will lead to substantial improvement of hand and finger strength in PD. This will need to be evaluated in future prospective studies. Both GripAble and Smart Sensor Egg appeared as suitable tools in PD, and this raises the question whether one of these two devices would be sufficient. The GripAble is a much bigger device, when compared to the Smart Sensor Egg, allowing to track and monitor hand movements reliably, such as hand pro- and supination, ulnar and radial deviation, and palmar flexion and extension. It also includes nine different games, each of them tackling these hand movements. Recently, it was shown that the GripAble can be used as an accurate and reliable digital grip strength dynamometer [33,39]. The Smart Sensor Egg has similar properties as the GripAble; however, the reliability of the sensor device still has to be evaluated. In addition, only one preliminary game is available at this point in time. The major advantage of this device is its pocket-sized design, making it easy to use in different locations. Further, the Smart Sensor Egg can be used as an object, and it can be manipulated between single fingers, allowing for coordinated finger training, which is not possible with the GripAble. Taking the advantages and disadvantages of both devices into account, choosing one device over the other may depend on whether the training focuses on hand or finger strength and personal preference of the user. However, future studies are needed in order to evaluate whether one device is superior to the other regarding hand-based training.

An interesting finding of our study was that higher usability ratings were associated with a higher flow state, which is in line with previous usability studies, which evaluated other types of game-based trainings [40,41]. The correlation of usability and flow fits also well with recent findings showing that mobile gaming satisfaction can be enhanced through a close integration of high usability and flow [42]. If the usability is high, it suggests that the new system helps guide users to achieve their own challengeable objectives [42]. This optimal challenge is then important in itself to move into a flow so that users receive an optimal gaming experience [18]. Thus, both concepts of usability and flow go hand in hand in order to achieve the best gaming experience [42]. GripAble and Smart Sensor Egg provide an optimal balance between challenge and skill acquisition when both healthy participants and persons with PD played the games. Adaptive challenge ability and continuous and self-reinforcing feedback is important for staying in the flow and for developing skills to master new challenges [18].

We did not find an association between cognitive function and flow. Previous research has, however, shown that higher levels of flow were related to higher levels of focused attention [43,44]. The reason why we could not find such an association might be due to the fact that we used the MoCA, which although being well-suited for PD [45], only contains a few items that assess attention. A more comprehensive cognitive evaluation could have detected more subtle attentional deficits, as it can be expected for persons with PD [5]. Another study, however, evaluated flow during a skill-based training [46], and it demonstrated that patients with brain trauma were still able to experience high flow, despite having attentional deficits. Brain-damaged patients may not depend on a fully intact attentional network, and they can still use other cognitive resources to move into a flow state. Which cognitive resources they can still draw on will need to be evaluated in future studies in persons with PD, and this question is outside the scope of this usability study. 

A limitation of this study is the small sample size, and care needs to be taken not to overinterpret the results. However, the numbers are in line with previous usability studies, which evaluated new gamed-based interventions for PD [25,47]. Moreover, we did not include persons with severe cognitive deficits; therefore, our findings cannot be generalised to persons with PD with severe cognitive impairment. It can be expected that persons with much more pronounced deficits may have higher usability needs, also requiring support from other persons when using new devices, especially at home. However, we did not find a relationship between demographic variables, such as disease severity, duration, and usability, suggesting that the devices may be well-suited for a broad population of persons with PD.

## 5. Conclusions

The present pilot study shows, for the first time, high usability of two new game sensor-based hand training devices, GripAble and Smart Sensor Egg, in healthy participants and persons with PD. The adaptable level of difficulty and continuous feedback during gameplay allows for sustained motivation and flow, which is crucial for a long-term improvement of hand function. Future prospective studies are needed, where both GripAble and Smart Sensor Egg are part of a comprehensive game sensor-based hand training for persons with PD.

## Figures and Tables

**Figure 1 sensors-22-06278-f001:**
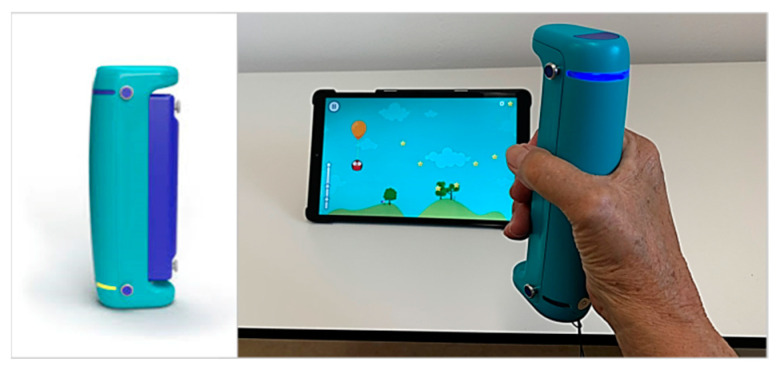
(**Left side**): GripAble. The device measures 55 mm × 35 mm × 170 mm and weighs 240 g. Right side: Participant playing Balloon Buddies with the GripAble (for more details see Ref. [33]).

**Figure 2 sensors-22-06278-f002:**
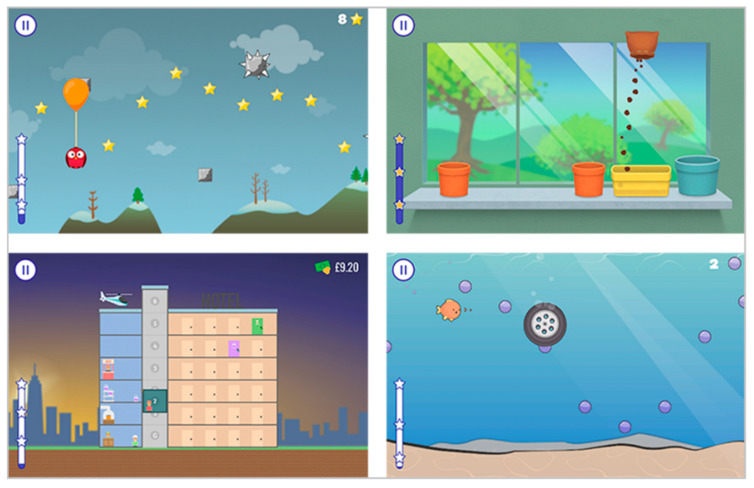
Games from the GripAble software that were used in the present study. (**Top left**): Balloon Buddies; (**top right**): Windowsill; (**bottom left**): Concierge; and (**bottom right**): Pufferfish.

**Figure 3 sensors-22-06278-f003:**
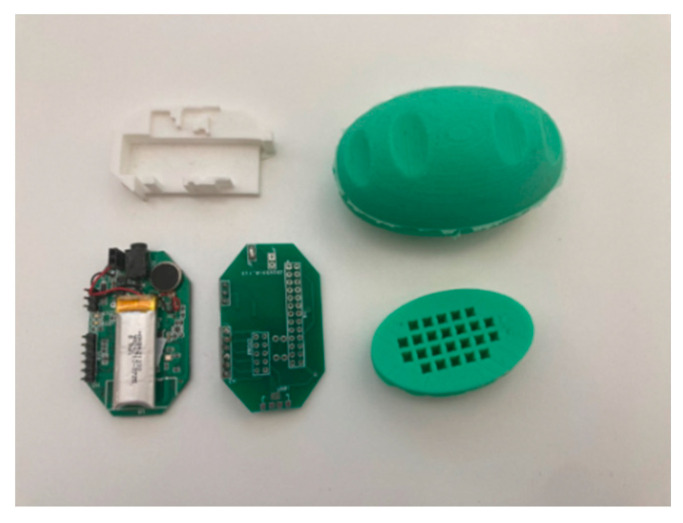
Smart Sensor Egg. The device is about 60 mm tall and weighs 50 g. (**Left side**): electronic board with an integrated pressure sensor and accelerometer. (**Right side**): silicone cover.

**Figure 4 sensors-22-06278-f004:**
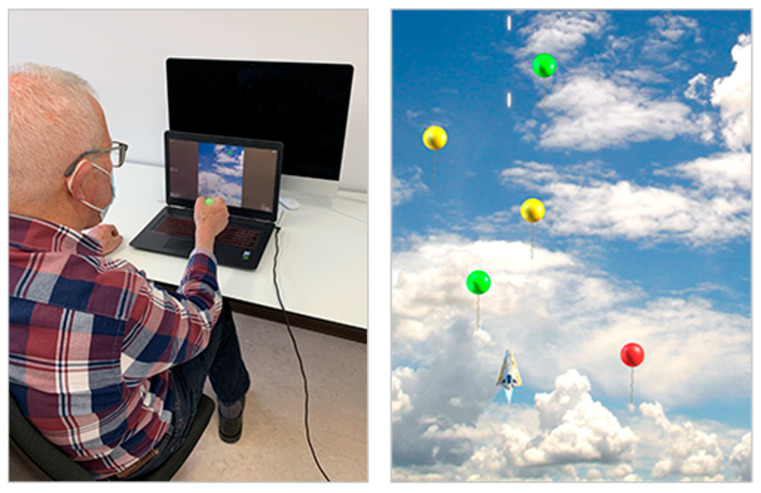
Participant playing Asteroid Shooter with the Smart Sensor Egg (**left**). Scenario of Asteroid Shooter game with the spaceship being controlled by the Smart Sensor Egg (**right**). The aim is to position the spaceship directly under the approaching targets (which are depicted as balloons in the lower difficulty levels; later levels feature asteroids as targets and include a space picture as the background image) and then to shoot them by squeezing the device.

**Figure 5 sensors-22-06278-f005:**
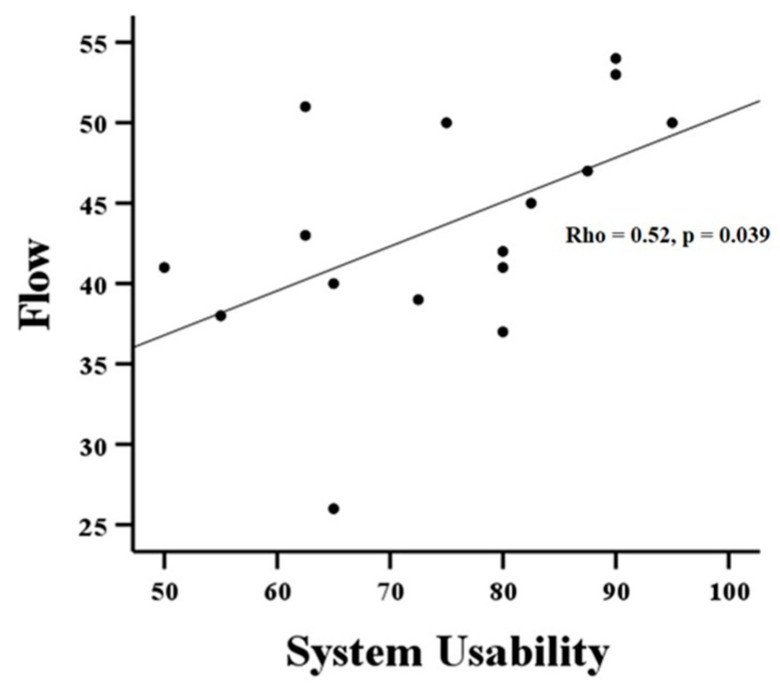
Correlation between flow and usability ratings. No significant correlations between SUS scores and disease duration (*r* = 0.07, *p* = 0.80) and MoCA scores (*r* = 0.16, *p* = 0.54) and the H and Y stage (eta = 0.327, *p* > 0.05) were found. Furthermore, the analysis revealed no significant correlations between flow and MoCA scores (*r* = 0.02, *p* = 0.90).

**Table 1 sensors-22-06278-t001:** Persons with Parkinson’s Disease (PD) and healthy participants (HS)—Clinical and demographic characteristics.

	PD (N = 8)	HS (N = 8)	*p*-Value
Age, y	63.5 (58–69.5)	63 (58–65)	0.710
Gender (m/f), n	6/2	2/6	0.046
MoCA	27.5 (25.25–27.5)	26.5 (23–28)	0.400
Handedness, (r/l), n	7/1	8/0	0.300
Disease duration, mo	31.5 (22.5–55)	-	-
Hoehn and Yahr stage	1.75 (1–2)	-	-

All values are presented as median and interquartile ranges (Q1–Q3 ranges) or otherwise stated; y = years; m = male, f = female; n = number of participants; MoCA = Montreal Cognitive Assessment; r = right, l = left; and mo = months.

## Data Availability

The datasets used and/or analysed during the current study are available from the corresponding author upon request.

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
