# Peer review of "Usability of Two New Interactive Game Sensor-Based Hand Training Devices in Parkinson’s Disease"

_sensors, 2022, doi:10.3390/s22166278_

Round 1

Reviewer 1 Report

The paper presents an assessment of two new interactive game training consoles for Parkinson Disease treatment.

The paper only addresses the usability aspect of the proposed consoles, each of them featuring one or multiple games, dedicated for various motions/joints. It is a good starting point, but it is unclear if these consoles are any better than the previously developed ones (like the traditional XBOX presented in: https://www.ncbi.nlm.nih.gov/pmc/articles/PMC5001816/). Can the authors comment on this aspect?

The analysis mainly includes users' feedback. This is important, of course, since the user acceptance is the starting point in most rehabilitation techniques. But no further indication is given regarding the measured efficiency of the devices. Do the authors plan an extensive study to assess this as well?

The authors state that: “Overall, we found  that  both  healthy  participants  as  well  as  persons  with  PD rated both new devices as  usable tools. No differences were found between the groups.” In what way did the healthy individuals rate these devices as “usable”?

Also, the authors have stated that the devices have been used by each individual 25 min. Have there been multiple sessions? How about the learning curve?

All in all, I believe this is a very early assessment of the two studied devices, which further needs to be developed to prove the real efficiency for the PD patients. By the way, in what measure would these be also suitable for other types of diseases as well?

Author Response

Please see thee attachment

Reviewer 2 Report

In this paper, the authors reported high usability of two game sensor-based hand training devices such as GripAble and Smart Sensor Egg in healthy as well as persons with PD. The study is useful with respect to Parkinson’s disease. However, the referee feels this manuscript is suitable for publication in this journal after some minor concerns.

Minor questions

1.     I suggest authors to change title as well as move results and discussion part after introduction

2.     Separate conclusion section should be included with major points.

3.     Minor types/language has been improved and figure 1 needs to be change, device in details/major parts must be provided in the image

Round 2

Reviewer 1 Report

The authors have addressed successfully all my questions and clarified the issues. I think the paper may be published.